



# Experimental study of the effect of a slat on the aerodynamic performance of a thick base airfoil

Axelle Viré[1], Bruce LeBlanc[1], Julia Steiner[1], and Nando Timmer[1]

[1]Wind Energy Section, Faculty of Aerospace Engineering, Delft University of Technology, Delft, The Netherlands

**Correspondence:** Axelle Viré (a.c.vire@tudelft.nl)

**Abstract.**

There is continuous effort to try and improve the aerodynamic performance of wind turbine blades. This experimental study focusses on the addition of a passive slat on a thick airfoil typically used in the inboard part of commercial wind turbine blades. Nine different slat configurations are considered, with both a clean and tripped main airfoil. The results are compared with the performances of the airfoil without slat, as well as the airfoil equipped with vortex generators. It is found that, when the airfoil is clean, the increase in lift-to-drag ratio due to the presence of a slat is larger than when vortex generators are used. This is also true for the tripped airfoil, but only at small angles of attack. As expected, in all configurations, the presence of the slat delays flow separation and stall. Finally, for a clean airfoil and small angles of attack, the slat decreases the lift-to-drag ratio of the main airfoil only. By contrast, as the angle of attack increases, it seems that the slat changes the flow field around the main airfoil in such a way that its lift-to-drag ratio becomes larger than for the airfoil without slat. These effects are less pronounced when the airfoil is tripped. This work helps to better understand the role of slat in improving the aerodynamics of blade sections. It can also be used to validate simulation tools in the field.

## 1 Introduction

The development of innovative add-ons, and their combination, are topics of high interest for wind turbine manufacturers. Such devices can increase the energy yield of a wind turbine by a few percent, leading to potentially significant reductions in levelised cost of energy. Add-on devices are already commonly used on commercial wind turbine blades. The type of add-ons, and their location along the blade, depend on the target objective which is often to improve the aerodynamic performances of the rotor. A good example of such devices are low-drag vortex generators (VGs), which are typically used at the inboard or mid-board sections of the blade. At these locations, the airfoil sections are rather thick with a high maximum lift in order to allow for the chord length to be reduced, without penalising greatly the overall energy yield. Thick airfoils inboard are also useful to reduce standstill loads under extreme conditions. However, the inboard part of the blade typically operates under large angles of attack. Hence, large flow separation may occur at these locations. In order to mitigate this, and increase the lift-to-drag ratio, vortex generators can be positioned in arrays in front of the separation line. These devices trigger the formation of small vortices in the boundary layer that re-energise the near-wall flow, hence preventing the flow to separate (Schubauer and Spangenberg, 1960). VGs are one type of wind turbine add-ons. Other types of devices are commonly used as well, either





to increase aerodynamic performance or to reduce trailing edge noise. Furthermore, add-ons can be either active or passive, depending on whether their characteristics change in time.

This work focusses on passive flow control devices that increase aerodynamic performance. In this context, aside from VGs, inboard devices include Gurney flaps, which increase the lift force on the blade section by increasing the effective camber
when placed on the pressure side and close to the trailing edge (Liebeck, 1978; Bach, 2016). Spoilers are another option and enhance the contribution of the lift and torque (Wentz, 1975). By contrast, add-ons can also be used outboard. This is the case of winglets, which are placed at the blade tip to reduce the downwash of the tip vortex, and therefore, decrease drag (Velte et al., 2016). A combination of multiple flow control devices can also be used on a given blade. An overview of this is given, for example, by Baldacchino (2019).

Although slats have been widely used in the aircraft industry, they are currently not used on commercial wind turbine blades. They could however be beneficial in the inboard part of the blade in order to delay the onset of stall, which is susceptible to occur due to the large angles of attack at these locations.The effect of a leading-edge slat on the flow field is however complex (Smith, 1975). Therefore, careful slat design and positioning are needed to ensure that the static stall angle of the main airfoil is increased, without overly increasing drag or causing a mixing between the wake of the slat and the boundary layer on the
main airfoil (Smith, 1975). In the last decade, the potential benefits of slats have also been investigated on thick airfoils. In this context, experimental studies report that the presence of a slat delays stall and increases the maximum lift coefficient (Pechli-vanoglou et al., 2010; Zahle et al., 2012). Interestingly, Zahle et al. (2012) further compared results from computational fluid dynamics (CFD) and experiments with an overall good agreement. However, the experimental measurements were polluted by three-dimensional wall effects at large angles of attack. Additionally, Jaume and Wild (2016) looked at a numerical shape
optimisation of both the slat and the main airfoil profile. The study showed that a combined optimisation leads to better aerodynamic performance than the superposition of both profiles optimised individually. Finally, Steiner et al. (2020) performed a parametric analysis using CFD simulations to optimise the design of a slat in combination with a thick airfoil at a Reynolds number of $Re = 10^7$. It was shown that a slat with a chord length equal to 30% of that of the main airfoil offers the aerodynamic benefit of a slat with a 40% chord length, without leading to a high positive pitching moment. Also, the increase in stall angle
and maximum glide ratio, due to the presence of the slat, was higher when the thickness of the main profile was increased.

The purpose of the present work is to explore experimentally the effectiveness of using a leading-edge slat on a thick base airfoil commonly used in commercial wind turbine blades. It also complements the numerical results obtained by Steiner et al. (2020) on a very similar setup. The paper is organised as follows. Section 2 describes the experimental setup, data acquisition, and post-processing methodology. Section 3 shows the results of the wind tunnel tests with a thick main airfoil and a variety
of configurations, namely: (i) without add-ons, (ii) with a leading-edge slat at two Reynolds numbers and a range of slat parameters, and (iii) with vortex generators. Finally, Section 4 summarises the main conclusions of this study.



## 2 Methodology

### 2.1 Experimental setup

The experiment is conducted in the low-speed low-turbulence tunnel (LTT) of Delft University of Technology[1]. This is an
atmospheric wind tunnel of the closed-throat single-return type, with interchangeable octagonal test sections with a length of
2.6 metres, a width of 1.80 metres and a height of 1.25 metres. The maximum Reynolds number for two-dimensional testing
is about 3.5 million. Here, the experiments are conducted at a Reynolds numbers of $Re = 1.5 \cdot 10^6$. The experimental setup
consists of the following components: the main airfoil model, the slat, and the attachment mechanism for testing different slat
configurations.

The main element is a composite DU00-W2-401 airfoil model with a chord length of $c_{\mathrm{main}} = 0.5m$. The model was mounted
vertically in the test section on two aluminium attachment plates flush with rotating discs in the upper and lower tunnel wall
that provide changes in angle of attack. The slat profile is a custom, cambered airfoil with a chord length equal to $0.3c_{\mathrm{main}}$,
as this was shown to lead to good performances in Steiner et al. (2020). The coordinates of the slat element are available
in LeBlanc et al. (2021). The slat is 3D-printed with an internal structure consisting of stiffening ribs and a hexagonal steel
rod, providing attachment to the aluminium plate and taking up the loads on the slat at high wind speeds. The slat surface
is smoothed with sandpaper and finished with spray paint. In this study, the streamwise position of the slat trailing-edge is
fixed at $s_{\mathrm{slat}} = 0.151c_{\mathrm{main}}$. The effect of two parameters are investigated, as shown in Fig.1: (i) the gap width $h_{\mathrm{slat}}$ between the
main airfoil and the slat trailing edge, and (ii) the slat angle $\beta_{\mathrm{slat}}$ relative to the main airfoil. In order to vary these parameters,
the steel rod has pre-manufactured set points in the mounting plate corresponding to pre-determined values of $h_{\mathrm{slat}}$ and $\beta_{\mathrm{slat}}$.
Figure 2 shows CAD views of the cross-section of the slat (left) and the end plate (right). The set of configurations investigated
here, and their associated labels, is presented in Table 1. The values of the slat angle $\beta_{\mathrm{slat}}$ are chosen based on previous works
(Steiner et al., 2020; Schramm et al., 2016; Jaume and Wild, 2016), which show that a slat angle of around $20°$ leads to the
best performance. In order to assess the sensitivity of the results to $\beta_{\mathrm{slat}}$, three different values separated by $5°$ are considered
here, namely $\beta_{\mathrm{slat}} = 16.4°, 21.4°, 26.4°$. It is also expected that reducing the gap width $h_{\mathrm{slat}}$, while avoiding confluent boundary
layers, increases the positive effect brought by the slat. However, the results obtained by Steiner et al. (2020) with a 2% gap
width (i.e. $h_{\mathrm{slat}}/c_{\mathrm{main}} = 0.02$) led to smaller lift coefficients, smaller lift-over-drag ratios, and a similar stall angle than with a
4% gap width. In this study, three values of the gap width varying between 2% and 4% are therefore investigated.

---

[1]https://www.tudelft.nl/en/ae/organisation/departments/aerodynamics-wind-energy-flight-performance-and-propulsion/facilities/low-speed-wind-tunnels/low-turbulence-tunnel/



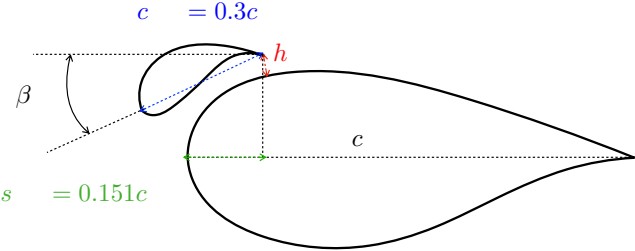

**Figure 1.** Illustration of the slat parameters of interest in this work: gap width $h_{\text{slat}}$ between the main airfoil and the slat, and (ii) the slat angle $\beta_{\text{slat}}$ relative to the main airfoil.

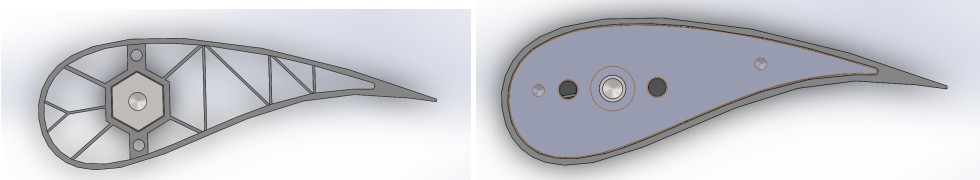

**Figure 2.** CAD design of the slat cross-section (left) and end plate (right).

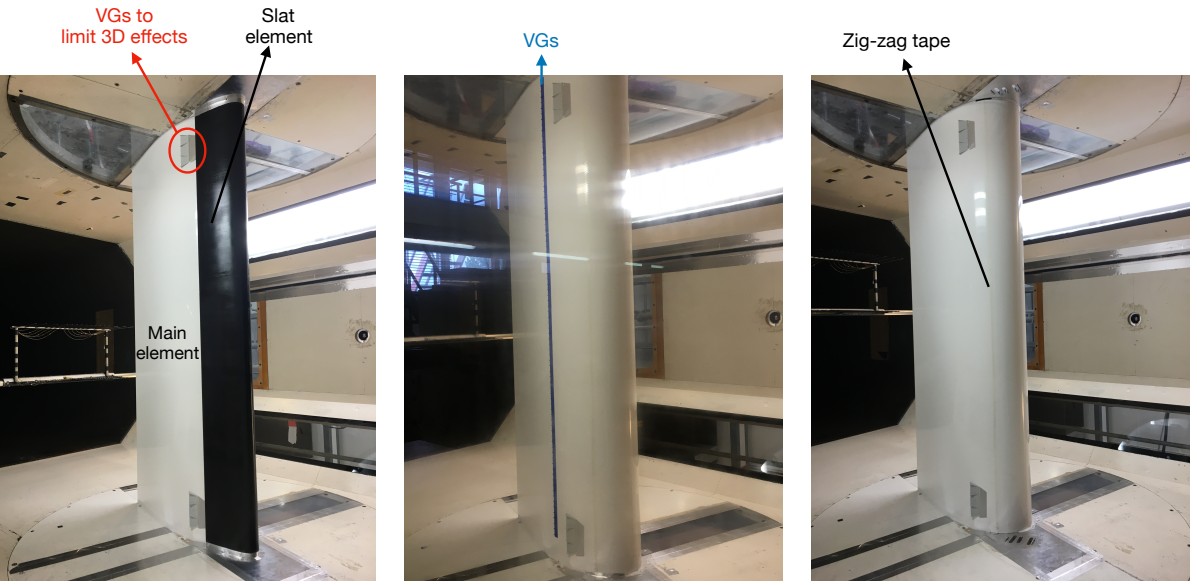

**Figure 3.** Photographs of the experimental setup: main profile with slat element (left), main profile with vortex generators (centre), main profile with zig-zag tape (right). In all the cases, VGs are used close to the walls to limit three-dimensional effects. This is highlighted in red on the left picture.





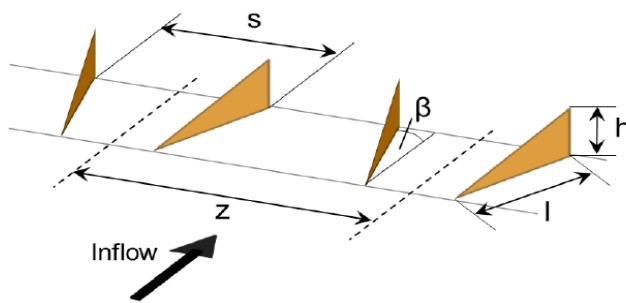

**Figure 4.** Illustration of the geometrical characteristics of the vortex generators.

**Table 1.** Set of configurations and associated labels.

| Configuration name | $h_{\mathrm{slat}}/c_{main}$ | $\beta_{\mathrm{slat}}$ [°] |
|---|---|---|
| A | 0.02 | 16.4 |
| B | 0.02 | 21.4 |
| C | 0.02 | 26.4 |
| D | 0.03 | 16.4 |
| E | 0.03 | 21.4 |
| F | 0.03 | 26.4 |
| G | 0.04 | 16.4 |
| H | 0.04 | 21.4 |
| I | 0.04 | 26.4 |

In order to mitigate interference effects due to the tunnel wall boundary layers, pairs of vortex generators (VGs) are installed on the main profile close to the walls, on both the pressure and suction sides of the main airfoil, as shown by Fig. 3 (showing the suction side only). As it will be shown in Section 3, this helps reducing three-dimensional flow effects on the measured pressure distributions. Since vortex generators are commonly used to improve the performances of wind turbine blades, this paper also presents results obtained when placing VGs on the main airfoil alone. In that case, the VGs are placed at $x/c_{\mathrm{main}} = 0.35$. The geometrical characteristics of the VGs are shown in Fig. 4 and the parameter values are as follows: $h/c_{\mathrm{main}} = 0.01$, $l/h = 2.2$, $z/h = 7.6$, $s/h = 3.2$, and $\beta = 15.8°$. The results obtained with the VGs alone are compared with the performances obtained with a slat.

Finally, all the tests are performed with both a clean and a tripped main airfoil. For the tripped cases, a zig-zag turbulator is placed on the main airfoil, as shown on the right photograph in Fig. 3. The details about the setup are given in Section 3.2.





## 2.2 Data acquisition and post-processing

The normal force coefficient $C'_n$ is obtained by integrating the pressure distribution from 94 and 11 pressure orifices in the
main element and the slat, respectively. For small angles, the measured lift coefficient $C'_l$ is expressed as

$$C'_l = \frac{C'_n}{\cos(\alpha)} - C'_d \tan(\alpha), \tag{1}$$

where $C'_d$ is the uncorrected drag coefficient measured with the wake rake and $\alpha$ is the angle of attack. The measured lift, drag
and moments coefficients are corrected for lift interference, solid and wake blockage, and wake buoyancy according to the
correction equation given by AGARDograph 336 (aga, 1998), while the pressure distributions are corrected according to the
method of Allen and Vincenti (1947), i.e.

$$C_l = C'_l \left(1 - t_1 - t_2 + t_3 - t_4\right), \tag{2}$$

where

$$t_1 = \frac{\sigma}{\beta^2}, \tag{3}$$

$$t_2 = 5.25\frac{\sigma^2}{\beta^4}, \tag{4}$$

$$t_3 = \frac{2 - M^2}{\beta^3}\Lambda\sigma\left(1 + \frac{1.1\beta c}{t}\right)\alpha^2, \tag{5}$$

$$t_4 = \frac{(2 - M^2)(1 + 0.4M^2)}{(4\beta^2)}\frac{c}{h}C'_d, \tag{6}$$

and $\sigma = \pi^2/48(c/h)^2$ is the wind tunnel blockage factor, $\beta = \sqrt{1 - M^2}$ is the compressibility factor, $M$ is the measured
apparent upstream Mach number, and the body shape factor is set to $\Lambda = 0.9087$. Primed coefficients denote uncorrected
values. Pre-stall, drag is measured using a wake rake placed behind the airfoil. The wake rake has 67 total pressure tubes
and 16 static pressure tubes. When the wake becomes unstable or wider than the wake rake, the latter cannot be used to
measure drag. Therefore, post-stall, the pressure lift and drag are used. The angle of attack at which the switch is made
between wake-rake drag and pressure drag is determined based on identifying flow separation from the measurements. These
angles also correspond to $C_l$ values that are slightly smaller than the maximum for a given case. The exact values are reported
in AppendixA. Independently of whether wake-rake or pressure drag is used, the resulting measured drag coefficient $C'_d$ is
corrected as

$$C_d = C'_d \left(1 - \Delta C_d - t_3 - t_4\right), \tag{7}$$

where the wake buoyancy correction is $\Delta C_d = 0$ when using the wake rake, and otherwise,

$$\Delta C_d = \Lambda\sigma\frac{1 + 0.4M^2}{\beta^3}\left(1 + \frac{1.1\beta c}{t}\alpha^2\right). \tag{8}$$



It must be noted that the present corrections are only valid until about an angle of attack of $20°$. In situations where significant

separation plays a role, these equations underestimate the effect of wake blockage and consequently may give too optimistic

lift coefficients. As this is mainly a comparative study, no effort has been undertaken to account for the additional blockage due

to separation. Data are recorded using an electronic data acquisition system and are on line reduced to show corrected force

and moment coefficients and pressure distributions. A thermal camera enables to visualise the location of flow transition along

with the pressure distribution. Wool tufts are also placed on the base airfoil to have a measure of flow separation and identify

possible three-dimensional wall effects during testing.

## 3   Results

### 3.1   Sensitivity to slat position

Figure 5 shows the lift coefficient $C_l$ as a function of the angle of attack $\alpha$ for different slat configurations, a clean main

airfoil, and at a Reynolds number of $Re = 1.5 \cdot 10^6$. The black dots represent the lift coefficient in the absence of a slat.

In that case, the maximum lift coefficient is reached at around $\alpha = 10°$ and remains close to unity as the angle of attack is

further increased. At $\alpha \approx 13°$, the lift coefficient presents a slight increase which is not expected. This feature did consistently

appear for multiple repetitions of the test, even after cleaning the main profile. Therefore, a possible reason for this small

disparity could be related to three-dimensional wall effects. Additionally, running the test at a slightly higher Reynolds number

($Re = 2 \cdot 10^6$) made this feature disappear. For $\alpha > 20°$, the main airfoil enters deep stall and the experimental results should

be disregarded. When a slat is added, the disparity around $\alpha \approx 13°$ also disappears and tuft visualisations demonstrate that the

flow is rather two-dimensional as expected (Fig. 8). The total maximum lift coefficient is significantly increased (factor up to

2.5) in comparison to the case without slat. For a given gap between the slat and the airfoil, the lift coefficient also increases

as the slat angle $\beta_{\text{slat}}$ decreases, except for the smallest gap and angle investigated, i.e. $h_{\text{slat}}/c_{main} = 0.02$ and $\beta_{\text{slat}} = 16.4°$,

which stalls already at around $\alpha \approx 15°$. The fact that stall is delayed at increasingly large slat angle is expected due to the

increase of effective cambering as $\beta_{\text{slat}}$ increases. Changing the slat gap width, for a given slat angle, does not significantly

change the lift coefficient. Similarly to Fig. 5, Fig. 6 shows the drag coefficient $C_d$ as a function of the angle of attack. It is

apparent that, for all the slat configurations, the total drag coefficient in the range $0° \leq \alpha \leq 7°$ is unaffected by the presence of

the slat. For larger angles of attack, the drag coefficient is reduced due to the presence of the slat. This observation might be

affected by the accuracy of the drag measurements, especially when the wake rake is used. However, these trends are in line

with the CFD results of Steiner et al. (2020). For completeness, Fig. 7 shows the ratio of lift-to-drag coefficients as a function

of $\alpha$. Two types of symbols are used: dots represent the data for the ensemble airfoil+slat, whilst crosses are the lift-to-drag

values integrated on the main airfoil only. It is clear that, for all the configurations considered here, the overall aerodynamic

performances of the ensemble airfoil+slat are improved in the presence of the slat, at least when $\alpha \geq 3°$. It is interesting to

note that, for $\alpha \approx 8°$, the lift-to-drag ratio computed on the main airfoil only (coloured crosses) is smaller than the lift-to-drag

ratio of the airfoil without slat (black dots). Thus, at small angles of attack, the increase in aerodynamic performance for the

slat configurations is largely due to the slat itself. In particular, the presence of a slat decreases the lift force on the main airfoil,



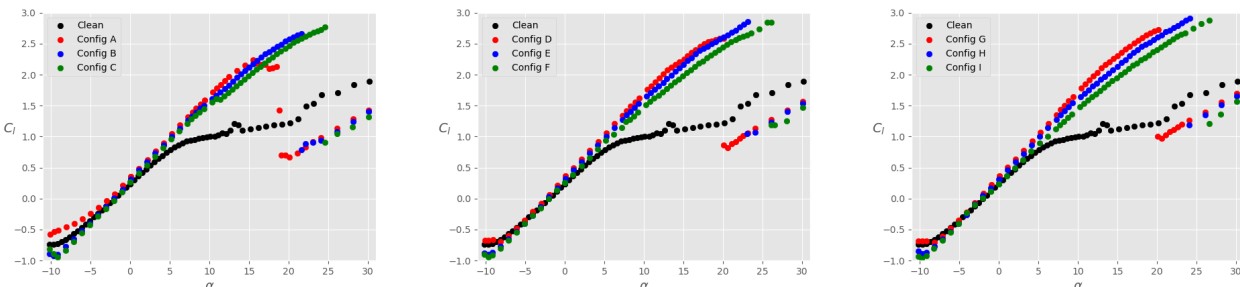

**Figure 5.** Total lift coefficient $C_l$ on the ensemble airfoil+slat as a function of the angle of attack $\alpha$, for different slat configurations, a clean main airfoil, and $Re = 1.5 \cdot 10^6$: $h_{\text{slat}}/c_{main} = 0.02$ (left), $h_{\text{slat}}/c_{main} = 0.03$ (center), $h_{\text{slat}}/c_{main} = 0.04$ (right). The black data are taken without slat, while the coloured data is in the presence of a slat with $\beta_{\text{slat}} = 16.4°$ (red), $\beta_{\text{slat}} = 21.4°$ (blue), $\beta_{\text{slat}} = 26.4°$ (green).

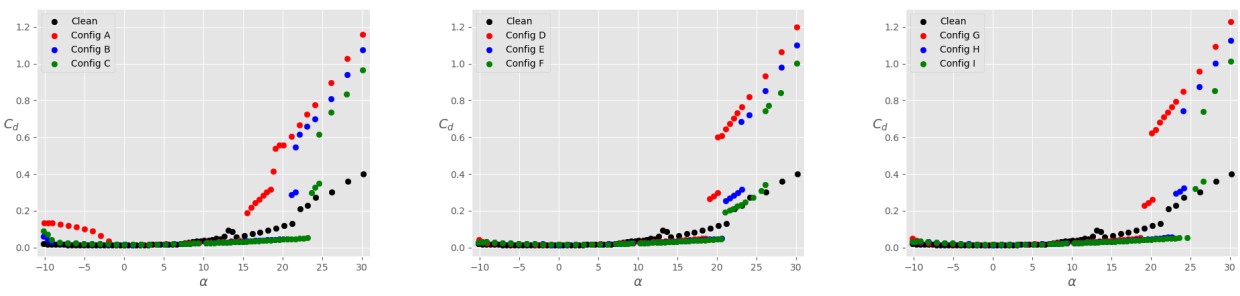

**Figure 6.** Total drag coefficient $C_d$ on the ensemble airfoil+slat as a function of the angle of attack $\alpha$, for different slat configurations, a clean main airfoil, and $Re = 1.5 \cdot 10^6$: $h_{\text{slat}}/c_{main} = 0.02$ (left), $h_{\text{slat}}/c_{main} = 0.03$ (center), $h_{\text{slat}}/c_{main} = 0.04$ (right). The black data are taken without slat, while the coloured data is in the presence of a slat with $\beta_{\text{slat}} = 16.4°$ (red), $\beta_{\text{slat}} = 21.4°$ (blue), $\beta_{\text{slat}} = 26.4°$ (green).

whilst it has a negligible effect on the drag force (not shown). This is in agreement with the description of Smith (1975). By contrast, at angles of attack larger than $8°$, the lift-to-drag ratio on the main airfoil alone (coloured crosses) is larger than that on the airfoil without slat (black dots). This means that the slat changes the flow field around the main airfoil in such a way

that the aerodynamic performance of the main airfoil itself is increased. As expected, this increase is further amplified when the contribution of the lift force on the slat is added (coloured dots). Again, the slat has a negligible effect on the drag force experienced by the main airfoil and the lift-to-drag ratios are consistent with those obtained by CFD simulations in similar conditions Steiner et al. (2020).

    Figure 9 shows the pressure distributions of the clean airfoil (left) and the airfoil with slat in case A (right) at an angle of

attack $\alpha = 10°$. It is clear that the pressure distribution on the main airfoil suction surface is significantly impacted by the presence of the slat. Due to the higher pressure on the slat lower surface, the low-pressure peak on the main airfoil is reduced,



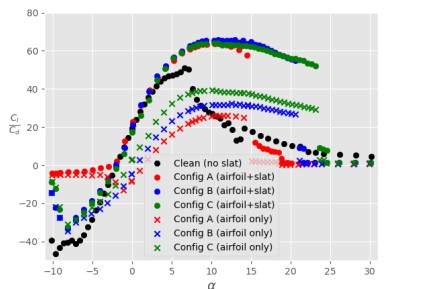 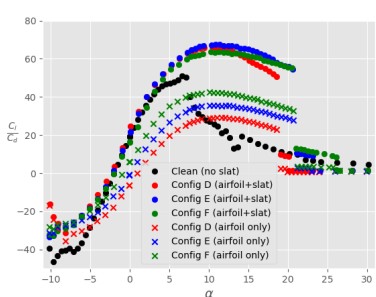 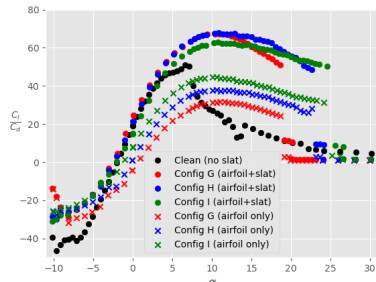

**Figure 7.** Lift-to-drag ratio $C_l/C_d$ as a function of the angle of attack $\alpha$ for different slat configurations, a clean main airfoil, and $Re = 1.5 \cdot 10^6$: $h_{slat}/c_{main} = 0.02$ (left), $h_{slat}/c_{main} = 0.03$ (center), $h_{slat}/c_{main} = 0.04$ (right). The black data are taken without slat, while the coloured data is in the presence of a slat with $\beta_{slat} = 16.4°$ (red), $\beta_{slat} = 21.4°$ (blue), $\beta_{slat} = 26.4°$ (green). Furthermore, dots represent the total lift coefficient (main airfoil + slat), whilst the crosses show the lift force on the main airfoil only.

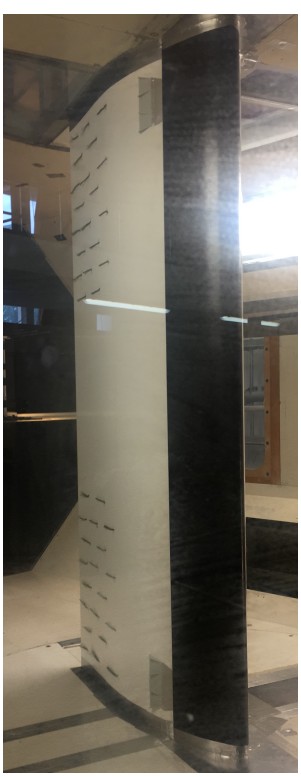

**Figure 8.** Tuft visualisation for a clean airfoil with a slat.





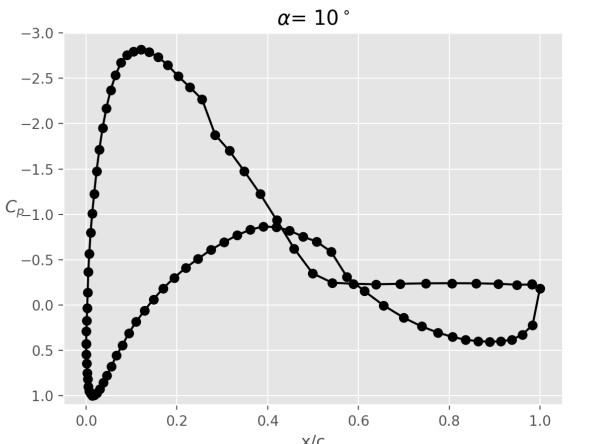
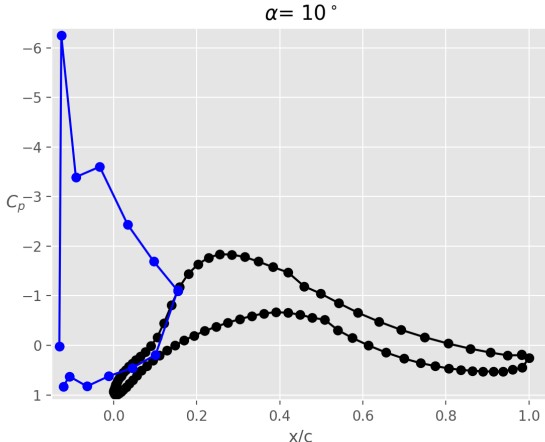

**Figure 9.** Pressure coefficient $C_p$ at $\alpha = 10°$ for a clean main airfoil: main airfoil alone (left), main airfoil with slat in case A (right), where both the pressure distributions on the main airfoil (black) and slat (blue) are shown.

also reducing the adverse pressure gradient towards the trailing edge. This enables the flow to stay attached up to the trailing edge, while the model without slat shows separation right from the mid-chord location.

This is mimicked by the associated infrared images presented in Fig. 10, with flow coming from the right. The model
without slat (left) shows a small laminar separation bubble followed by transition indicated by the thin dark band at about one-third of the chord. Separation may be hard to see in this picture. It is indicated by the lightest grey area. Both transition and separation locations may have been slightly influenced by the presence of the pressure orifices located in the lower part of the picture. Transition on the slat can be clearly seen in the right infrared picture, indicated by the dark band between the dashed lines. Its location coincides with the kink in the pressure distribution after the pressure peak on the slat leading edge.
Due to the low Reynolds number of the slat, transition will almost certainly be realised through a small laminar separation bubble. It is followed by turbulent attached flow. Although the leading edge of the airfoil is not visible due to the presence of the slat, combining the pressure distribution and the infrared image shows that the main airfoil upper surface is laminar up to the mid-chord position, followed by an attached turbulent boundary layer.

### 3.2  Sensitivity to roughness

In order to assess the sensitivity of the results to surface roughness, the boundary layer on the main airfoil is tripped using a zig-zag turbulator tape placed at the 10% chord location on both the pressure and suction sides. The tape height is calculated according to Braslow and Knox (1958) using a critical Reynolds number of 400. This leads to a tape of thickness $0.2mm$. Furthermore, the tape has a width of $6mm$ and an angle of $30°$. Figure 11 shows the lift coefficient $C_l$ as a function of the angle of attack $\alpha$ for different slat configurations. The black dots again represent the lift coefficient in the absence of slat.
As expected, but still highly undesirable, tripping the boundary layer significantly reduces the lift coefficient at low angles of





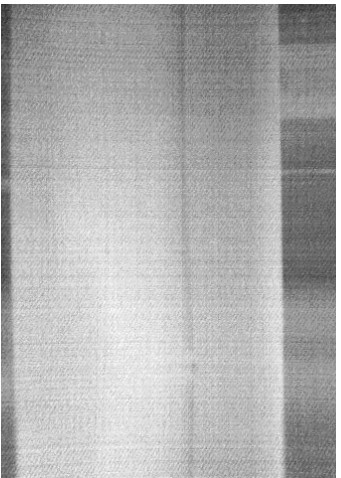 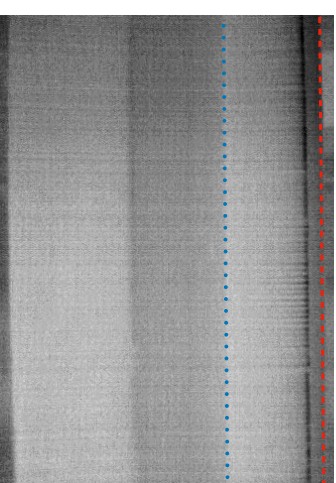

**Figure 10.** Infrared images of the suction side of the clean main airfoil at an angle of attack $\alpha = 10°$: without slat (left) and with slat under configuration A (right). In the right figure, the red dashed line indicates the slat leading edge, whilst the blue dotted line shows the slat trailing edge. Flow is going from right to left.

attack. The negative effect of the zig-zag tape can be alleviated by using vortex generators, as shown in Fig. 12 for the main airfoil without slat. Note that the small abnormality in the lift coefficient at $\alpha \approx 13°$ for the clean case is not observed in the tripped case (with and without VGs). When the slat is mounted on the main airfoil, the zig-zag tape also leads to a loss of $C_l$ at small angles of attack, although this effect is less pronounced than in the absence of slat (Figure 11). This observation is also

made on the lift contribution coming from the main airfoil (not shown). At negative angles of attack, the lift coefficient on the ensemble airfoil+slat is almost identical to that of the airfoil without slat. Consequently, the presence of the slat decreases the contribution of the lift force coming from the main airfoil. This is not the case for positive angles of attack. In particular, for $\alpha > 5°$, the slat has no influence on the lift force experienced by the main airfoil only, as that contribution is identical to the lift force on the main airfoil without slat (not shown). This explains why the lift coefficient on the ensemble airfoil+slat is then

larger than that of the airfoil without slat (Fig. 11). Additionally, similar trends than with the clean airfoil are obtained, namely an overall increase in $C_l$ in the presence of a slat, with larger $C_l$ as $\beta_{\mathrm{slat}}$ decreases. Also, there is a small decrease of $C_l$ as $h_{\mathrm{slat}}$ increases, although this is very small.

The drag coefficients are presented in Fig. 13. As expected, drag is larger in the tripped case compared to the clean results. However, the presence of the slat slightly decreases $C_d$ at all positive angles of attack before stall. As for the results on the

195 clean airfoil, the contribution of the slat to the total drag force is very small. The positive effect that the slat has on the overall drag reduction is therefore mostly due to the fact that the slat changes the flow field around the main airfoil. Figure 14 shows the lift-to-drag ratio for all the tripped configurations. As already mentioned before, the overall aerodynamic performance of the tripped system is smaller than in the clean case. The addition of a slat has a positive influence on the overall aerodynamic





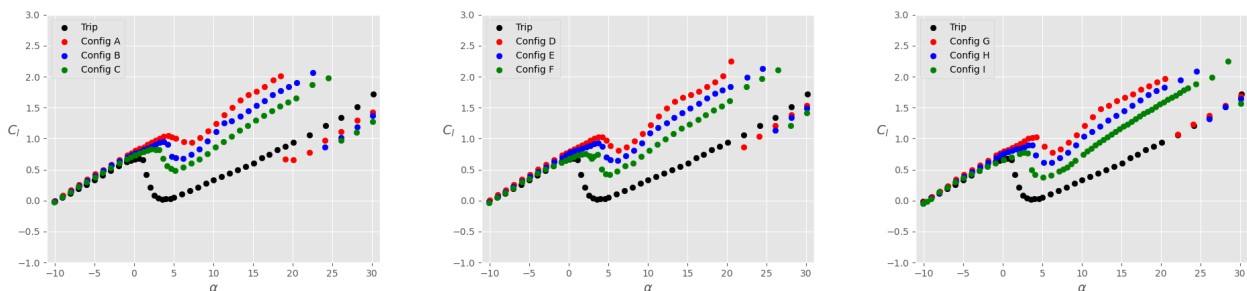

**Figure 11.** Total lift coefficient $C_l$ on the ensemble airfoil+slat as a function of the angle of attack $\alpha$, for different slat configurations, a tripped main airfoil, and $Re = 1.5 \cdot 10^6$: $h_{slat}/c_{main} = 0.02$ (left), $h_{slat}/c_{main} = 0.03$ (center), $h_{slat}/c_{main} = 0.04$ (right). The black data are taken without slat, while the coloured data is in the presence of a slat with $\beta_{slat} = 16.4°$ (red), $\beta_{slat} = 21.4°$ (blue), $\beta_{slat} = 26.4°$ (green).

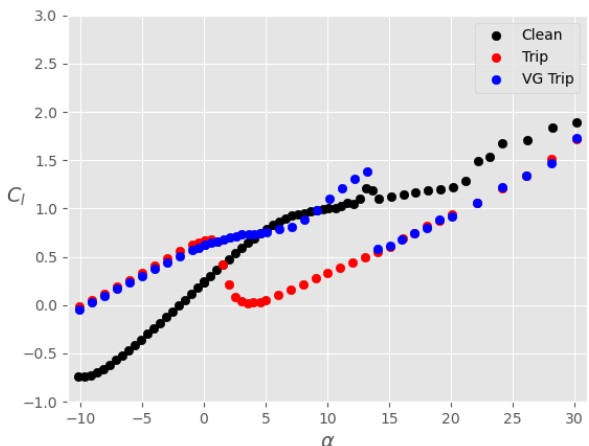

**Figure 12.** Lift coefficient $C_l$ as a function of the angle of attack $\alpha$ for a main element without slat: clean airfoil without VGs (black), tripped airfoil without VGs (red), tripped airfoil with VGs (blue).

performance. However, as opposed to the clean cases, the ratio $C_l/C_d$ computed on the airfoil alone (in the presence of a slat)
is rather similar to that of the airfoil without slat (not shown here for the sake of graph clarity).

Figure 15 shows infrared images of the flow past the airfoil at $\alpha = 10°$, without (left) and with (right) slat. In the right figure, the red dashed line indicates the slat leading edge, whilst the blue dotted line shows the slat trailing edge. It can be seen that, in both cases, the flow is rather two-dimensional as expected. Also, the presence of the slat leads to a more uniform flow on the main airfoil. This is also apparent on the pressure coefficient shown in Fig. 16 for the same conditions.





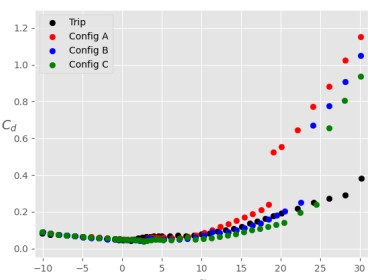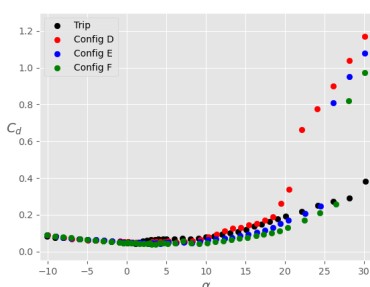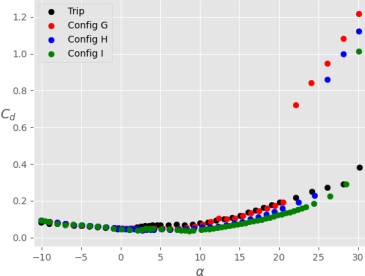

**Figure 13.** Total drag coefficient $C_d$ on the ensemble airfoil+slat as a function of the angle of attack $\alpha$ for different slat configurations, a tripped main airfoil, and $Re = 1.5 \cdot 10^6$: $h_{\mathrm{slat}}/c_{main} = 0.02$ (left), $h_{\mathrm{slat}}/c_{main} = 0.03$ (center), $h_{\mathrm{slat}}/c_{main} = 0.04$ (right). The black data are taken without slat, while the coloured data is in the presence of a slat with $\beta_{\mathrm{slat}} = 16.4°$ (red), $\beta_{\mathrm{slat}} = 21.4°$ (blue), $\beta_{\mathrm{slat}} = 26.4°$ (green).

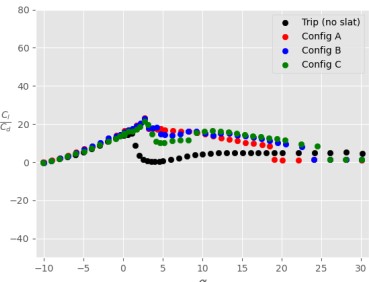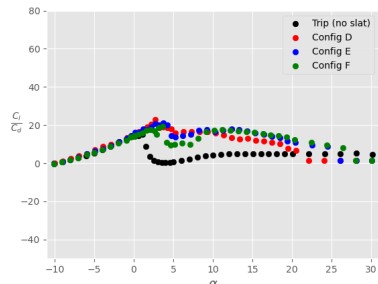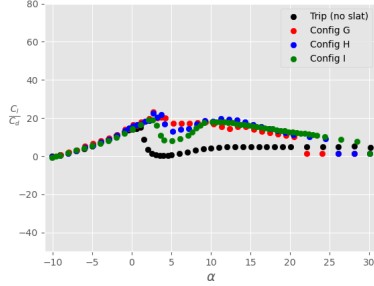

**Figure 14.** Lift-to-drag ratio $C_l/C_d$ on the ensemble airfoil+slat as a function of the angle of attack $\alpha$ for different slat configurations, a tripped main airfoil, and $Re = 1.5 \cdot 10^6$: $h_{\mathrm{slat}}/c_{main} = 0.02$ (left), $h_{\mathrm{slat}}/c_{main} = 0.03$ (center), $h_{\mathrm{slat}}/c_{main} = 0.04$ (right). The black data are taken without slat, while the coloured data is in the presence of a slat with $\beta_{\mathrm{slat}} = 16.4°$ (red), $\beta_{\mathrm{slat}} = 21.4°$ (blue), $\beta_{\mathrm{slat}} = 26.4°$ (green).

## 3.3 Comparison of the performances with VGs

Figures 17 and 18 show the polars for the lift and drag coefficients, respectively, when the main airfoil is equipped with VGs instead of a slat (red dots). For comparison, results obtained with the slat under configuration A are also shown in these plots (blue dots). Again, both clean (left) and tripped (right) conditions are shown. In both conditions, it is clear that the lift coefficient is considerably increased due to the presence of the VGs, which suppress separation. The increase is however smaller than when a slat is used. Furthermore, at small positive angles of attack, the lift-to-drag ratio is reduced in the presence of the VGs in the clean case (Fig. 19, left), which is not the case with a slat. The aerodynamic performances obtained with VGs are larger than with the slat only in the tripped conditions and for $6° < \alpha < 14°$, where the small gain in $C_l$ brought by the slat is not enough to compensate for the small associated increase in $C_d$. Outside of this range of angles of attack, the slat seems to be aerodynamically more advantageous than VGs, at least for the conditions investigated here. Of course, the slat brings additional challenges for the structural design of the blade, which are not taken into account in this study.



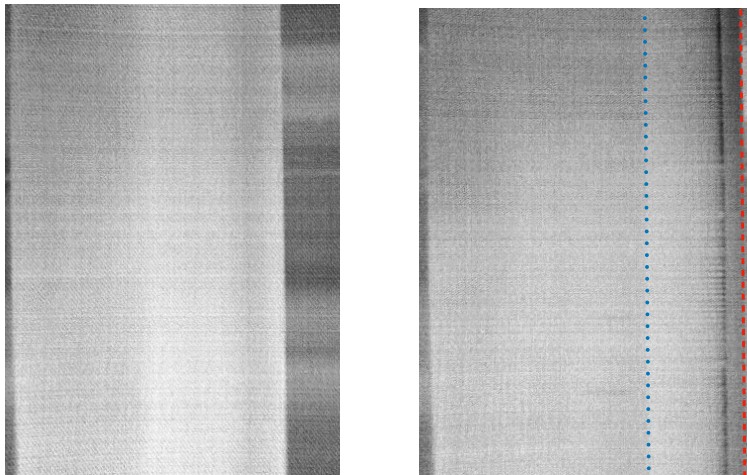

**Figure 15.** Infrared images of the suction side of the tripped main airfoil at an angle of attack $\alpha = 10°$: without slat (left) and with slat under configuration A (right). In the right figure, the red dashed line indicates the slat leading edge, whilst the blue dotted line shows the slat trailing edge.

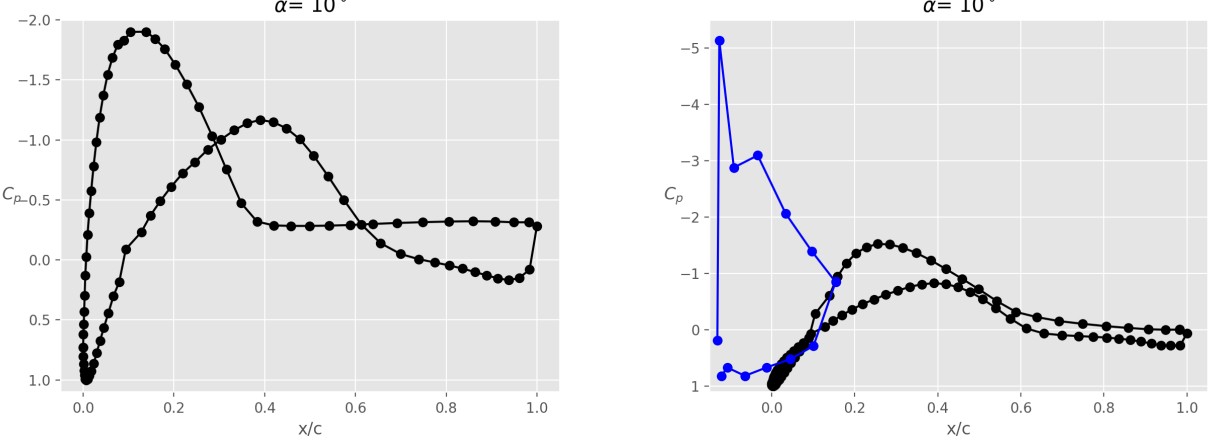

**Figure 16.** Pressure coefficient $C_p$ at $\alpha = 10°$ for a tripped main airfoil: main airfoil alone (left), main airfoil with slat in case A (right), where both the pressure distributions on the main airfoil (black) and slat (blue) are shown.




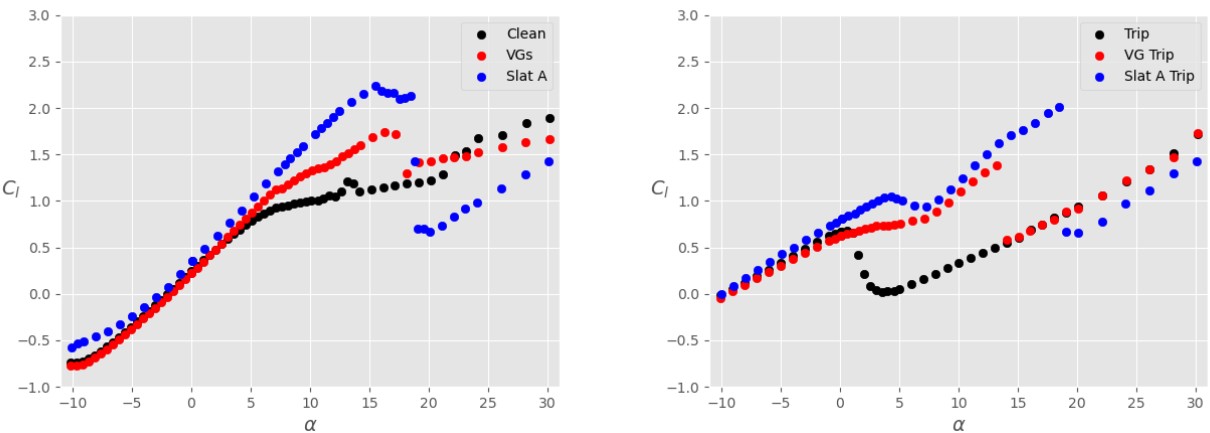

**Figure 17.** Lift coefficient $C_l$ as a function of the angle of attack $\alpha$ for the main airfoil at $Re = 1.5 \cdot 10^6$, without VGs and without slat (black), with VGs only (red), and with slat only in configuration A (blue). Left: clean airfoil; right: tripped airfoil.

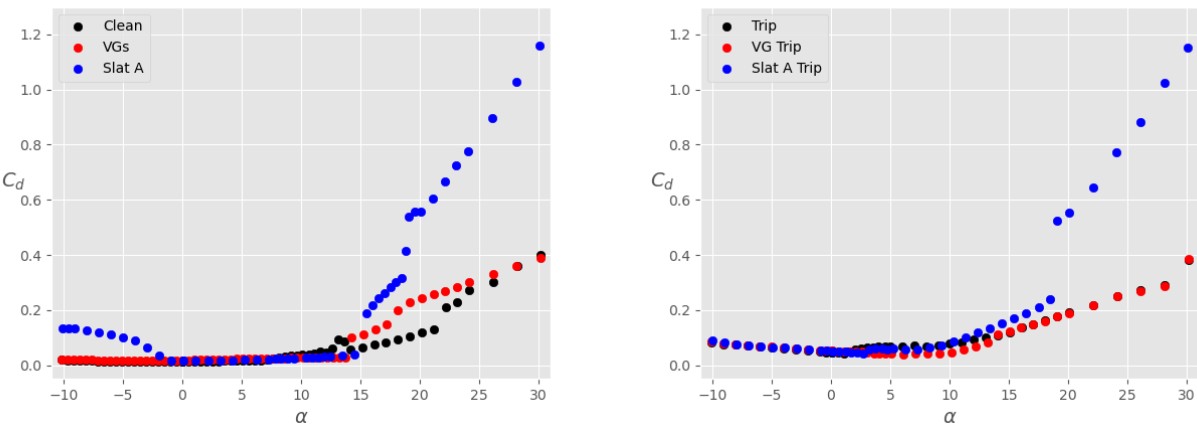

**Figure 18.** Drag coefficient $C_d$ as a function of the angle of attack $\alpha$ for the main airfoil at $Re = 1.5 \cdot 10^6$, without VGs and without slat (black), with VGs only (red), and with slat only in configuration A (blue). Left: clean airfoil; right: tripped airfoil.

## 4  Conclusions

This paper summarises the main results obtained from wind tunnel experiments on a DU00-W2-401 airfoil, equipped with either a slat or vortex generators, in both clean and tripped conditions. The results suggest that the use of a slat can significantly increase the aerodynamic performance of the system. For a clean airfoil and small angles of attack, the presence of a slat

decreases the lift-to-drag ratio of the main airfoil only. This is in line with the slat effect described in the literature (Smith,





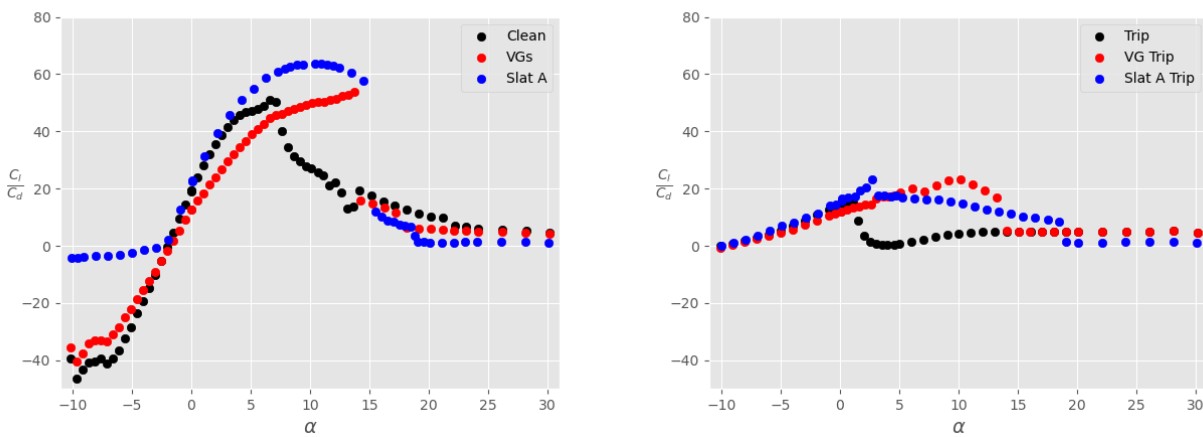

**Figure 19.** Lift-to-drag ratio $C_l/C_d$ as a function of the angle of attack $\alpha$ for the main airfoil at $Re = 1.5 \cdot 10^6$, without VGs and without slat (black), with VGs only (red), and with slat only in configuration A (blue). Left: clean airfoil; right: tripped airfoil.

1975). However, because of the positive lift on the slat itself, the overall lift-to-drag ratio of the ensemble airfoil+slat is either equal to, or larger than, that of the airfoil without slat. As the angle of attack increases beyond approximately $7°$ to $10°$, the presence of a slat modifies the flow field around the main airfoil in such a way that the aerodynamic performance of the latter alone is increased. The overall lift-to-drag ratio is therefore largely better than without using a slat. This is in line with

the results from CFD simulations obtained in similar conditions (Steiner et al., 2020). The slat also clearly delays stall, as documented in the literature. The present results show some dependencies of the lift coefficient to the position and angular orientation of the slat. In particular, for the range of parameters investigated here, larger slat angles usually lead to smaller values of $C_l$. Increasing the gap between the slat and the main airfoil can lead to slightly better values of $C_l$, which suggests that there is an optimal gap between the slat and the airfoil, beyond which the performance will likely decrease again. When

the airfoil is tripped, the presence of a slat is beneficial at all positive angles of attack and does not change the aerodynamic performances at negative angles of attack, compared to the case of an isolated airfoil. It is also shown that the use of a slat can partly alleviate the loss of lift at low angles of attack under tripped conditions.

      The present conclusions hold for the geometries and parameters investigated here and do not consider any structural challenges that arise when attaching a slat to a wind turbine blade. In particular, the effect of the slat on the overall blade mass and

aeroelastic responses, as well as the logistics of attaching the slat to the blade, are aspects that would need further analysis. Future work should therefore focus on incorporating these aspects in the overall assessment of the feasibility and potential of using slats on commercial wind turbine blades.





## Appendix A:  Values of the angles of attack delimiting the use of either the wake-rake drag or the pressure drag

Table A1 lists, for each case presented in this paper, the angle of attack below which the wake rake drag is used for the

calculation of $C_d$.

**Table A1.** Values of angles of attack below which the wake-rake drag is used.

| Airfoil condition | VG present | Slat configuration | $\alpha$ [°] |
|---|---|---|---|
| Clean | No | None | 8 |
| Clean | Yes | None | 14 |
| Clean | No | A | 15 |
| Clean | No | B | 21 |
| Clean | No | C | 23.5 |
| Clean | No | D | 19 |
| Clean | No | E | 21 |
| Clean | No | F | 21 |
| Clean | No | G | 19 |
| Clean | No | H | 23 |
| Clean | No | I | 25 |
| Trip | No | None | 1.5 |
| Trip | Yes | None | 3 |
| Trip | No | A | 3 |
| Trip | No | B | 3 |
| Trip | No | C | 3 |
| Trip | No | D | 3 |
| Trip | No | E | 3 |
| Trip | No | F | 3 |
| Trip | No | G | 3 |
| Trip | No | H | 3 |
| Trip | No | I | 3 |

**Acknowledgements**

The authors would like to acknowledge Rijksdienst voor Ondernemend Nederland (RVO) through the TSE Hernieuwbare En-
ergie funding scheme (ABIBA project). We are also grateful for the technical support of Stefan Bernardy and Emiel Langendijk
of the Low-speed Laboratory in helping with the design and preparation of the model setup. We also would like to thank Wind



Tunnel Services. Finally Nicholas Balaresque of Deutsche WindGuard is acknowledged for giving us the opportunity to use
their DU00-W2-401 airfoil model as basis for the measurements.



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

# Author contribution

AV: original manuscript, post-processing and interpretation of results, funding acquisition, responsible supervisor. BL: scientist in charge of the experiment (preparation, run, post-processing), interpretation of results, manuscript revisions. JS: support with model design and parameters, interpretation of results, manuscript revisions. NT: support to prepare the experiment, interpretation of results, responsible of the wind tunnel facility.