# Peer review of "Experimental study of the effect of a slat on the aerodynamic performance of a thick base airfoil"

_Wind Energy Science, 2021_

## Author Comment (AC1)

**Reply to Reviewer 1**

The manuscript describes a well setup experimental investigation on the impact of a slat device on a relatively thick wind turbine airfoil. The conclusions are well founded and agree with current results obtained by other working groups. The state of konwledge is well captured and reported.

*Thank you very much for your thorough review and valuable feedback. We have addressed your points, as indicated below in colour.*

Nevertheless, there are a few topics that need to be addressed, which are outlined according to their appearance in the text.

page 5, table 1: The setup of the different cases for slat positions is hard to remember thruoughout the manuscript. It is suggested to rearrange the table for clarity:

columns; gap

rows: deflection

entries: config letters

This would make it clearer afterwards to see, which configs have same angle and which have same gap

*Thanks for the suggestion, Table 1 is changed accordingly.*

page 5, line 83f: Can you give any details on the geometric properties of these vortex generators as they are different from those listed a few lines down. How do they compare to other experiments of 2D wings

*The geometrical characteristics of the VGs used to mitigate wall effects are now added to the text.*

page 5, line 87ff: Is there a refenece on the design of these values for the VGs - as they differ from those reported by Godard & Stanislas (2006, https://www.sciencedirect.com/science/article/pii/S127096380500163X) to be optimal. Godard & Stanislas also linked the optimal height to the boundary layer thickness, not to the wing chord. So, what are the assumptions on the boundary layer thickness for selecting the VG parameters.

*The VGs used here are in line with those used on a commercial blade within the project that funded this research. That design was developed by FlowChange/We4Ce, based on windtunnel results and production technology.*

page 6, line 94: the number of pressuere ports seems quite low number for pressure integration, especially on the slat nad espcecially for drag (in case where the wake traverse data gets doubtful). Did you check the pure integration error.

*We have computed the Cp distributions with Xfoil and compared integrating Cp on the whole profile against integrating Cp from the data at the 13 pressure tabs only (we have corrected in the text that we*

*used 13 tabs instead of 11). The integration error is about 11%. We have added to the manuscript as follows: "Also, the low number of pressure tabs on the slat leads to an integration error of about 11\%. This is quantified by comparing the integration of $C_p$ values from Xfoil along all the slat coordinates with the integration considering only the data at the locations of the pressure tabs."*

page 6, line 99f: The correction of Allen & Vincenti (NACA TR-782, 1944) is not a correction of the pressure distribution but a correction of the lift and moment coefficients as well as the effective incidence. They can be applied to the integrated values from the uncorrected pressure distribution. But the drag term is often omitted (see AGARD-AG336) especially for 2D airfoil measurmments since measurment inaccuracy of the drag coefficient can have a high impact on the corrected values of lift and pitching moment coefficient. The pressure distribution should be only corrected by teh blockage/wake-blockage corrections to take into account inaccuracies in static pressure and flow velocity.

*We have rephrased part of the text in Section 2.2 on data post-processing. Regarding the corrections from Allen and Vincenti (A&V) in NACA TR-782, they do give correction equations both for the uncorrected force and moment coefficients and for the associated pressure distribution (if the force and Cm coefficients come from integration of the cp-distribution). The reason that we do not mention using the standard A&V correction on these coefficients is the fact that a 2nd order term is added for the lift interference (impacting the corrected lift, moment and angle of attack) in the form of t2, which indeed comes from AG109 instead of AG336 mentioned in the text. Then again, the fact that we omit the wake-buoyancy correction term to correct the wake rake drag coefficient is based on a discussion by Rogers in AG336.*

*In addition, we do make use of the formulation of A&V for the correction of the pressure distribution, which is slightly different from the more straightforward method of*

*Cp=(Cp'±ΔCp)q'/q in which ΔCp is the lift-interference effect.*

*A&V argue that the contributions of solid blockage and lift interference should be treated separately; a correction for the blockage due to the base profile and a correction for streamline curvature applied to the lift per unit chord. This leads to slightly different formulations for the Cp's on the airfoil upper and lower surfaces. In the context of this paper those differences are of no importance.*

page 6, eqs. 2-6. Please be precise to the referenceing of the corrections. Not all terms are found in AG-336 and NACA TR-782 (e.g. terms t1 and t2), and most likely origin from the older AGARD AG-109. c, h and t are not explained.

*See our explanation above. The definition of c, h, t are added.*

page 7, line 134: Incase results should be disregarded since they are definitely questionable, it would be better to not show them at all.

*All the figures of Cl, Cd, and Cl/Cd have been re-generated, omitting the data points at alpha>20deg that cannot be trusted.*

page 7, line 136: referecne to fig 8 is out of sequence (decribed before Fig.7)

*The former Fig 8 has been moved up in the manuscript so that it is now Fig 6 and is in sequence with the text.*

page7, line 138: "except" in which sense? The mentioned increase of stall angle and lift coefficient is also there for this case - or is it the 2.5 increase factor?

*The sentence is re-written to avoid confusion. The mentioned increase of lift coefficient with decreasing beta is indeed also apparent for Config A. We just wanted to point out that Config A stalls at a smaller angle of attack, which obviously leads to a lower Cl beyond the stall angle of attack (alpha=15deg).*

page 7, line 139f: please explain, why cambering delays the stall? Usually, it doesn't increase the angle of attack where stall occurs, but the lift coefficient at same incidence.

*We have removed this sentence as we observe this only for 1 case (going from config A to config B) and, as the Reviewer pointed out, this can probably not be generalized.*

page 7, line 140f: please be precise whether you discuss the lift coefficent at same incidence or the maximum lift coefficient. Anyhow, the mentioned comparison is hardly observed in Fig.5

*It is now specified that we mean the lift coefficient at a given angle of attack.*

page 7, line 142ff: sure? or is it due to the scale - the high values in the separated regime make the differences in the low alpha range hardly visible. In fact, one would expect a drag increase in the low incidence range with the slat due to the additional friction losses. And the drag reduction at higher incidenes is simply by suppressing the separation!

*Indeed when zooming in, we can see that Cd with a slat is slightly larger than without a slat. This is now corrected in the manuscript. Also, we clarify that the drag is increased at all incidences, but the benefits of the slat are visible when looking at the lift-to-drag ratio.*

page 8, figure 6, The scaling of the drag coefficient is emphasizing the differences in the regime where the fllow is separated (and where drag measurements are questionable). It would be better to change the scale to lower CD values (e.g up to CD=0.2) highlighting the differences in the attached flow regime, which are not visible due to closeness of values and size of symbols

*The scaling of all Cd plots is changed to maximum Cd=0.6. Going lower would cut some data points at large angle of attack. Also, the maximum angle of attack shown is now limited to alpha=20deg. We hope that this improves clarity.*

page 8, line 158: typo: insert "by" ahead of "Steiner"

*This is corrected.*

page 8, line 161: This is a misleading argumentation. Since this is a channel, of course the pressure on both sides is similar. But it is not the high pressure on the slat lower side but the slat circulation that reduces the pressure level on the main wing (see Smith (1975) )

*This is clarified, thanks.*

page 11, figure 10:  as a hint for future research: It is hard to argue on the boundary layer state without a clear reference. Thus, it is usual to place an artificial dustrubance somewhere in the laminar region to see the color difference of laminar and turbulent wall temperature

*Thanks for the hint!*

page 11, line 190:  Be careul with the naming "clean airfoil" - decide whether it means "swithout tripping" or "without slat"

*Clean airfoil means without tripping throughout the manuscript. This is now clarified at several places earlier in the manuscript.*

page 12, line 201:  not "past" - or if yes, please tell how, you get IR images of the free air flow.

*This is replaced by "on the airfoil".*

page 14, figure 16:  A direct comparison to the data in fig 9 in one graph would help to see and explain the differences.

In fact, as the lower side is that much affected, this seems to be more than just an effect of laminar/turbulent transisiont, but there is a risk of over-tripping. In this view, the reference used is dealing wiht dstributed roughness elements, while the used method is a zig-zag-tape. It my be likely (and supported by the reults, that the tripping is too thick for only removing the risk o the lamaminar seperation bubble, see e.g. AIAA-1997-0511. All in all, you should decide, whether you really want to open up this question in this context. In fact, to verify that the tripping wa appropriate, a more detailed study on different trippings would be needed. to exclude "overtripping".

*We think that the Cp curves are still interesting to show and we kept separate figures for clarity. We do mention that an analysis on different tripping methods has not been performed and could be useful for future studies. There is of course a risk of overtripping. However, since the combination and airfoil and slat is considered for wind turbine blades, we do not consider this as a problem.*

page 16, line 228f:  There is definitely an optimum slat position, wee Woodward & Lean, 1993 (AGARD-CP-515) !

*Good point, we rephrased and added the reference.*

References:

Reference ot AGARD AG-336:  in case of summary report the edoir shall be named:

Ewald, B.F.R. (ed.), Wind Tunnel Wall Correction, AGARDograph 336, AGARD, 1998.

Reference to Allen & Vincenti: The Report NACA TR 782 is of 1944.

*This is corrected.*

Reference to "Jaume":  the last name of the first author is "Manso Jaume" - or only "Manso" (it's Spanish, thus two last names are common) - please also correct in the text (page 2, line 44).

*This is corrected in the bibtex entry.*

---

## Author Comment (AC2)

**Reply to Reviewer 2**

**General Comments**

This papers describes the effect of a slat on the aerodynamic performance of a thick airfoil commonly used for wind turbine rotors near the hub. The experimental results include lift and drag, pressure distributions and flow visualization.

The paper is well written and the scientific quality is very good. I have some general remarks:

*Thank you very much for your thorough review and valuable feedback. We have addressed your points, as indicated below in colour.*

I would like to see a comparison of the pressure distribution and lift and drag values of the baseline airfoil with data from the literature. This would help to validate the experimental setup.

*We have modified Figs 5, 7, 8 to include the results of Xfoil and compare with the experimental Cl and Cd curves for the clean main airfoil alone. Note that Xfoil is expected to be reliable in the linear part of the Cl polar. Hopefully this gives some confidence in the experimental setup.*

The pressure distribution on the slat is measured with only 11 pressure ports which is a low resolution. Did you make an assessment on the possible error on the lift computation?

*We have computed the Cp distributions with Xfoil and compared integrating Cp on the whole profile against integrating Cp from the data at the 13 pressure tabs only (we have corrected in the text that we used 13 tabs instead of 11). The integration error is about 11%. We have added to the manuscript as follows: "Also, the low number of pressure tabs on the slat leads to an integration error of about 11\%. This is quantified by comparing the integration of $C_p$ values from Xfoil along all the slat coordinates with the integration considering only the data at the locations of the pressure tabs."*

Section conclusions: I miss some recommendations based on the observations on when and how to apply slats on airfoils (guidlines).

*This is clarified by summarizing the best slat angle and gap width as found in the study.*

**Specific comments**

69: I would insert a reference to Fig 2. here and highlight the hexagonal rod in Fig 2.

*Done.*

Fig 2: It would be helpful to include the locations of the 11 pressure ports in the figure in order to understand the pressure distribution.

*The location of the pressure tabs (corrected to 13) is indicated by red dots in Fig 2 (left).*

87: Based on which knowledge was this position chosen: x/c_main=0.35?

*This location is chosen as a compromise between increasing lift coefficient and reducing the associated drag penalty. This is now added to the text. Some internal confidential tests (company-related) confirmed that this location is good but we cannot provide more information on this unfortunately.*

134 & Fig 5,6 etc: As you state that for alpha > 20° the results should be disregarded, I suggest to eliminate these data from the plots in the related figures because they are misleading and you would increase the resolution in the plots.

*All the figures of Cl, Cd, and Cl/Cd have been re-generated, omitting the data points at alpha>20deg that cannot be trusted. Plot resolutions have also been adapted.*

132: „Therefore, a possible reason for this small disparity could be related to three-dimensional wall effects" Did you observe this with the tufts? I'm asking because later you state „tuft visualisations demonstrate that the tuft visualisations demonstrate that the flow is rather two-dimensional as expected (Fig. 8)" which implies that you observed some three-dimensionality…

*The tuft visualizations indicated that the flow is rather 2D. We checked this up to alpha=11deg. Unfortunately, we have not checked the tuft visualization at exactly the alpha=13deg condition, which is the only case for which we have this disparity. We know this does not come from the airfoil surface and that this observation was repeatable. However, we could not pinpoint exactly the cause of it.*

143: "For larger angles of attack, the drag coefficient is reduced due to the presence of the slat". I see that the slat configuration represented by the red dots has a larger drag.

*Indeed this was a mistake in the text and is now corrected. The drag should indeed be larger in the presence of the slat.*

Fig. 8: Which flow conditions are used here? Re, AoA etc…

*The photo was taken at Re=1.5*10^6. Unfortunately, we have not noted the exact AOA. We have modified the figure title with: "Example of tuft visualisation obtained for a clean airfoil (i.e. no tripping) with a slat at $Re=1.5\cdot 10^6$ and a small angle of attack."*

204: I have a hard time recognizing this statement in Fig 15: "Also, the presence of the slat leads to a more uniform flow on the main airfoil". What do you mean by "uniform flow"?

*This is rephrased as "Also, the presence of the slat alleviates flow separation on the main airfoil"*

**Technical corrections: typing errors, etc.**

109: comma after "Pre-stall" is not needed

*This is rephrased as "The drag coefficient pre-stall…" to increase clarity.*

111: "Therefore, post-stall, the pressure lift and drag are used" This sentence reads strange.

*This is rephrased as "Therefore, post-stall, the pressure measurements on the airfoil and slat surfaces are used instead."*